# A Dual Representation Framework for Robot Learning with Human Guidance

**Ruohan Zhang\*[1], Dhruva Bansal\*[1], Yilun Hao\*[1], Ayano Hiranaka[2], Jialu Gao[3], Chen Wang[1],**
**Roberto Martín-Martín[4], Li Fei-Fei[1,5], Jiajun Wu[1,5]**
[1]Department of Computer Science, Stanford University
[2]Department of Mechnical Engineering, Stanford University
[3]Tsinghua University
[4]Department of Computer Science, The University of Texas at Austin
[5]Institute for Human-Centered AI (HAI), Stanford University

**Abstract:** The ability to interactively learn skills from human guidance and adjust behavior according to human preference is crucial to accelerating robot learning. But human guidance is an expensive resource, calling for methods that can learn efficiently. In this work, we argue that learning is more efficient if the agent is equipped with a high-level, symbolic representation. We propose a dual representation framework for robot learning from human guidance. The dual representation used by the robotic agent includes one for learning a sensorimotor control policy, and the other, in the form of a symbolic scene graph, for encoding the task-relevant information that motivates human input. We propose two novel learning algorithms based on this framework for learning from human evaluative feedback and from preference. In five continuous control tasks in simulation and in the real world, we demonstrate that our algorithms lead to significant improvement in task performance and learning speed. Additionally, these algorithms require less human effort and are qualitatively preferred by users. Project website: https://sites.google.com/view/dr-hrl.

**Keywords:** Human Guidance, Evaluative Feedback, Preference Learning

## 1 Introduction

Human guidance refers to a diverse set of human training signals provided to a learning agent [1–5]. These alternative forms of human input can be combined with the conventional reward signal in reinforcement learning (RL) [6] or demonstrations in imitation learning (IL) [7–9] to facilitate learning. Recently, the robot learning community has increased its attention to human guidance as a mechanism to overcome two critical challenges: 1) the low sample efficiency of learning algorithms, and 2) the effort in manually specifying the objectives for learning. Human guidance is helpful for these challenges because 1) guidance like evaluative feedback [10–27] can be used as domain knowledge to speed up learning, and 2) through their guidance, humans can define the learning objectives for robots so that the learning algorithm better infers and aligns to the underlying human goals and values, such as their preferences [28–46].

Despite its benefits, human guidance is a scarce and valuable resource, and human-in-the-loop mechanisms strive to find ways to reduce the amount of guidance required from humans to make them more broadly and practically applicable. This is only possible if the human guidance is properly represented and interpreted [47–49]. The first step is to choose a representation that allows the learning agent to query humans more efficiently and learn with less human guidance [47–49].

Robots typically use a fine-grained state and action space for continuous control. However, when humans observe, evaluate, and guide robot behaviors, their representation is likely different. For example, in a continuous control task such as placing an object, the robotic agent typically has access to low-level states including proprioceptive information and other objects' poses. But the guidance

---

\*indicates equal contribution; correspondence to zharu@stanford.edu

provided by humans is typically based on higher-level abstract information, such as *is the robot's end-effector to the left or the right of the goal?*, or *is the robot grasping the object?* This observation leads to the "dual representation hypothesis" proposed in this work, inspired by cognitive science studies [50–53]. This is analogical to the "fast and slow" systems proposed by Kahneman [52], in which the fast system manages intuitive, automatic, unconscious behaviors while the slow system manages logical, calculating, conscious thoughts [52]. We hypothesize that the "slow" system and its representation are useful in guiding learning agents.

In this work, we propose a *dual representation* framework for robot learning from human guidance: the robotic learning agent uses a low-level state representation for learning control policies, but keeps a *symbolic scene graph* [54–57] as a high-level representation of human internal states. We show that this framework enables novel learning algorithms: Dual Representation–based Evaluative Feedback (DREF) and Dual Representation–based Preference Learning (DRPL). DREF is based on the idea of *uncertainty-aware active learning* that allows the agent to estimate the uncertainty of human feedback in unseen states. Such generalization ability is achieved by using scene graph representation to group low-level states into abstract states. DRPL builds upon *scene graph–based trajectory segmentation and selection*, allows efficient reward learning from chosen trajectory segments.

In three simulation tasks and two real-robot tasks, we demonstrate that our proposed approaches lead to significant improvements in learning speed and performance. For challenging long-horizon real-world robot manipulation tasks, we show that DREF can learn to solve the task efficiently, while end-to-end RL algorithms fail to solve because of the high-dimensional continuous state and action space. Critically, we observe a significant reduction in the amount of human guidance required for learning, and an improvement in overall user experience, making learning from human guidance methods more practical and appealing for real robot learning.

## 2 Related Work

Among the multiple forms of human-in-the-loop robot learning [58–60], in recent years, the robotics community has paid increased attention to human guidance [3] because of being powerful and easy to collect, and complements standard training signals such as rewards or demonstrations.

**Learning from human evaluative feedback.** This is an approach in which human trainers monitor the learning process of an agents, and provide a scalar signal to indicate whether the observed behavior is desirable [10–27]. The agent then learns a policy to maximize positive feedback from humans. This approach has the advantage of placing minimum demand on both the human trainer's expertise and the ability to provide guidance, compared to learning from demonstrations. Significantly, human evaluation is often interpreted as a value function [14, 16] or an advantage function [15, 17], not as the reward itself. Nonetheless, human evaluation can be naturally combined with environment rewards so the agent learns simultaneously from both sources [18–20]. Evaluative feedback often targets individual state-action pairs. Hence one outstanding challenge is to generalize observed human feedback to unseen state-action pairs.

**Learning from human preference.** In this framework, the learning agent queries human trainers for their preferences over a set of exhibited behavior trajectories [28–46]. Preferences can be used to directly learn policies [28, 29], a preference model [30], or a reward function [31, 32]. Recent works often learn a hypothesized latent human reward function from the preference signals, and combine preference learning with deep RL to learn a policy afterwards [34–37]. Notably, a recent work attempts to align agent representation with human representation in preference learning [61]. While evaluative feedback targets a state-action pair, preference learning targets trajectories. Selecting optimal query trajectory length is a challenging problem in preference learning [3], we will show that the dual representation learning framework provides an intuitive way to segment lengthy trajectories.

**Scene graph as abstract representation.** In robot learning, abstract representation has been an active research topic, exemplified by recent works in neuro-symbolic robot learning [62–65]. Human-robot interaction tasks, such as shared autonomy [66, 67], highlight the importance of abstract representation since humans often communicate high-level goals to the agent in this setting. Low-dimensional, abstract representations could potentially be learned [68–72] but sample efficiency is a major limiting factor when applying these methods to physical robot learning from real humans.

A scene graph, often specified by humans, is a form of abstract representation for state information in which objects are represented as nodes and relations between objects are represented as edges [54–

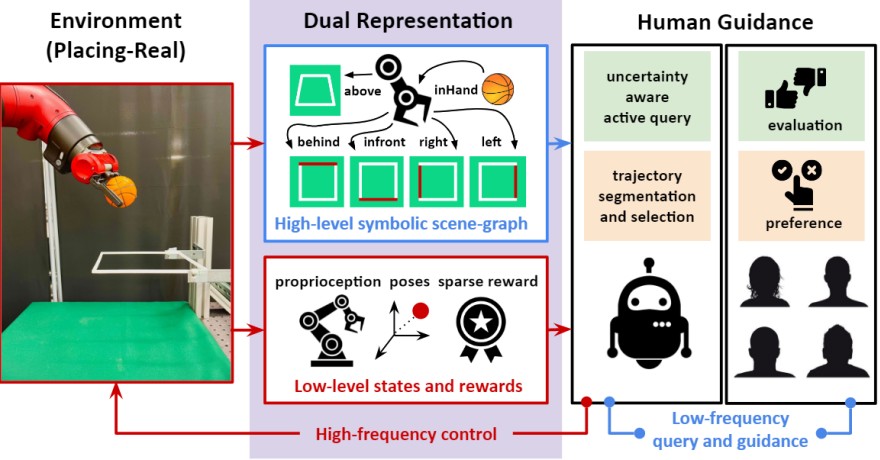

Figure 1: Overview of the proposed dual representation framework. The robot maintains two state representations. The first one is for learning a fine-grained, low-level continuous control policy. The second representation is specified by an expert human in the format of a symbolic scene graph. The robot uses this representation to actively query human trainers for evaluation or preference during the training process.

57]. Scene graphs can store explicitly and compactly information about object geometry, placement, semantics, and relationships, making them suitable for tasks that require sophisticated reasoning about these types of information. Recently, scene graphs have started to be used in robotics [73–75] as a state representation that facilitates planning and reasoning. We explore how a symbolic scene graph could be a useful additional representation for human-in-the-loop robot learning. We hypothesize that it allows the learning agent to query humans more efficiently and learn with less human guidance than using the low-level, raw state representation alone.

## 3  Method

Our method is designed to overcome a significant challenge in human-in-the-loop robot learning: human feedback is expensive, and frequently asking for guidance is infeasible in real-world robotic systems. To address this challenge, we propose a novel dual representation to facilitate human-in-the-loop policy learning. Below we introduce the general human guidance learning problem and our proposed dual representation setups (Sec. 3.1). Then, we propose an algorithm to learn from evaluation feedback that decreases the amount of human feedback needed during policy learning (Sec. 3.2), and a query generation algorithm to efficiently query humans for their preference (Sec. 3.3).

### 3.1  Dual Representation for Learning from Human Guidance

We represent the robot learning problem as a Markov Decision Process denoted by the tuple $\langle \mathcal{S}, \mathcal{A}, P, R, \gamma \rangle$, where $\mathcal{S}$ is the state space, $\mathcal{A}$ is the action space, $R : S \times A \rightarrow \mathbb{R}$ is the reward function, $P : \mathcal{S} \times \mathcal{A} \times \mathcal{S} \rightarrow [0, 1]$ is the transition function, and $\gamma$ is the discount factor. A policy $\pi : \mathcal{S} \times \mathcal{A} \rightarrow [0, 1]$ is a mapping from $\mathcal{S}$ to probability distribution over $\mathcal{A}$.

We propose a dual representation framework for human-in-the-loop robot learning that consists of the common low-level state information from the environment and an additional high-level symbolic scene graph $\mathcal{G}$. $\mathcal{G}$ is represented as a binary vector of $dim(\mathcal{G})$, where each dimension represents a unary state of an object or a pairwise semantic relation between the objects and the robot. Let $g$ be an instance of $\mathcal{G}$, i.e., an abstract state which contains infinite low-level, continuous states. The objects and relations in $\mathcal{G}$ are specified by humans based on task knowledge. The proposed dual representation framework is shown in Fig. 1.

### 3.2  Learning from Evaluative Feedback with Dual Representation

Human evaluative feedback contains rich task-level knowledge that could be used to assist robot learning. We adopt an *active* learning setting, in which an RL agent *asks* humans for their evaluative feedback. Hypothetically, asking and receiving feedback in the right state could lead to better learning results with less human effort, but it is unclear *when* to ask for feedback. Our key insight

---

**Algorithm 1** Dual Representation–based Evaluative Feedback (DREF)

---

1: Initialize $\mathtt{UCB1}(g), \forall g \in \mathcal{G}$, as 0
2: Initialize $\mathcal{D}$ as the replay buffer
3: Initialize network weights for actor $\theta_A$, environment critic $\theta_E$, and human feedback critic $\theta_H$
4: **for** t = 1, T **do**
5:     Select $a_t = \pi(\cdot|s_t; \theta_A)$
6:     Infer current scene graph state $g_t$ from $s_t$
7:     **if** $rank(\mathtt{UCB1}(g_t)) \leq k$, across $g \in \mathcal{G}$ sorted by UCB1 score **then**
8:         Query for evaluative feedback $H_t(s_t, a_t)$
9:     **end if**
10:    Execute $a$, observe reward $r$, and next state $s_{t+1}$,
11:    Store transition $(s_t, a_t, r_t, H_t, s_{t+1})$ in $\mathcal{D}$
12:    Sample random minibatch of transitions $(s_t, a_t, r_t, H_t, s_{t+1})$ from $\mathcal{D}$
13:    Perform a gradient step according to Equations 3 4, and 5 for $\theta_H, \theta_E, \theta_A$.
14:    Update $\mathtt{UCB1}(g), \forall g \in \mathcal{G}$ using Equations 1,2
15: **end for**

---

is that querying for human feedback can be formulated as a multi-armed bandit problem with symbolic scene graph representation. This formulation brings the opportunities to use formal methods designed for discrete state and action spaces, such as Upper Confidence Bound (UCB1). Integrating UCB1 with continuous control enables efficient *uncertainty-aware active learning* from humans.

TAMER+RL [18–20] is a widely used framework for learning from evaluative feedback. For the RL part, we use Soft Actor-Critic (SAC) [76]. In addition to the environment reward, human trainers provide a scalar signal $H_t(s, a) \in \{-1, 0, +1\}$ to indicate whether the observed state-action pair is desirable or not. Our goal is to reduce the amount of total feedback while achieving the same or better task performance.

Inspired by the standard UCB1 for value function in bandit problems, we use the following equation to estimate the upper confidence bound of human feedback prediction error (FPE), the type of uncertainty we care about, in an abstract state $g_t$:

$$\mathtt{UCB1}(g_t) = \mathtt{FPE}(g_t) + c\sqrt{\frac{2 \log N_t}{N_t(g_t)}}, \tag{1}$$

where $N_t$ is the total number of human feedback received at time $t$, and $N_t(g_t)$ is the number of feedback given to the abstract state $g_t$. The constant $c$ weighs the exploitation and exploration terms. $\mathtt{FPE}(g_t)$ is the average feedback prediction error for all the low-level states encountered so far that belongs to $g_t$:

$$\mathtt{FPE}(g_t) = \frac{1}{N_t(g_t)} \sum_{i=0}^{t} \mathbb{1}(s_i \in g_t) \|\hat{H}(s_i, a_i) - H(s_i, a_i)\|_2^2. \tag{2}$$

We can estimate $\hat{H}(s, a)$ using an additional critic head in SAC. Assume $\theta_H, \theta_E, \theta_A$, parameterize the human feedback critic, the environment critic, and the actor, respectively, the learning objectives are:

$$\mathcal{L}(\theta_H) = \mathbb{E}_{(s,a,H)\sim\mathcal{D}} \|\hat{H}(s, a; \theta_H) - H(s, a)\|_2^2 \tag{3}$$

$$\mathcal{L}_{\mathtt{SAC}}(\theta_E) = \mathbb{E}_{(s,a)\sim\mathcal{D}} \|Q(s, a; \theta_E) - \hat{Q}(s, a)\|_2^2 \tag{4}$$

$$\mathcal{L}(\theta_A) = \mathbb{E}_{s\sim\mathcal{D}} \left[ \mathbb{E}_{a\sim\pi(\cdot|s;\theta_A)} \left[ \alpha \log(\pi(a|s; \theta_A)) - (Q(s, a; \theta_E) + \lambda\hat{H}(s, a; \theta_H)) \right] \right] \tag{5}$$

In Eq. 4, $\mathcal{L}_{\mathtt{SAC}}(\theta_E)$ is the standard soft Bellman residual in SAC [76, 77]. In Eq. 5, $\alpha$ is the temperature parameter in SAC [76, 77]. Note that the actor updates the policy distribution in the direction suggested by the weighted average of both critics. The agent learns a policy to maximize expected positive feedback from humans and environment reward simultaneously.

The full algorithm, **DREF** or Dual Representation–based Evaluative Feedback, is shown in Algorithm 1. We first calculate the running mean of UCB value for each abstract state and rank them.

---

**Algorithm 2** Dual Representation–based Preference Learning (DRPL)

---

1: Collect a set of random trajectories $\mathcal{T}$
2: Set prior of reward weights $p^0(\theta)$ randomly
3: Segment all trajectories in $\mathcal{T}$ based on abstract state changes: $\mathcal{T}_{seg} = \{\xi_1, \xi_2, ...\}$
4: **for** $i = 0, dim(\mathcal{G})$ **do**
5:     Select two segments $\xi, \xi' \in \mathcal{T}_{seg}$, with ties broken arbitrarily
       s.t. their associated abstract states $g[i] \neq g'[i]$
       **and** $d(g, g') = 1$ **or** $d(g, g') = \min_{g,g'} d(g, g')$
6:     Query for preference $q(\xi, \xi')$
7:     Update posterior $p^{(i+1)}(\theta) \propto P(q|\theta)p^i(\theta)$, where $P(q|\theta)$ is the probability of preference
       response $q$ given $\theta$
8: **end for**

---

Then we query for feedback at a specific state $s_t$ if its abstract state $g_t$ has a high rank (determined by a hyperparameter $k$). As a result, the agent actively asks for feedback in abstract states with uncertainty in the human evaluation of actions, e.g., query for feedback when the gripper and the ball move into the hoop for the first time, as shown in Fig. 1. With the low-level state and action space alone, this cannot be easily done because the number of states is infinite. The abstract state representation effectively groups these states together, and estimates the uncertainty in predicting feedback of a new state using the average FPE of other states in its group. We hypothesize that in this way, the dual representation could lead to efficient *uncertainty-aware active learning* from human feedback, which we will demonstrate in our experiments.

### 3.3 Preference Learning with Dual Representation

Preference learning is an important method to define the objective for learning agents. While evaluative feedback targets a state-action pair, in preference learning human trainers indicate their preference over a pair of trajectories, from which the agent learns a reward function $R$. For simplicity, we assume that reward function $R$ is a linear combination of state-action features [78]: $r(s_t, a_t) = \theta^T \phi(s_t, a_t)$, hence the goal is to infer $\theta$. Our goal is to reduce the amount of human guidance: the algorithm should accurately estimate $\theta$ with a minimum number of queries.

Here we address two issues in preference learning: *generating* and *selecting* meaningful queries. For query generation, selecting trajectory length is challenging [3]. Indicating preference over longer trajectories requires cognitive effort (e.g., summing all the rewards in each trajectory). Short trajectories (e.g., random 1-2 seconds clips [34]) allow humans to provide feedback of high granularity, but these clips may not be meaningful or comparable. After trajectories are segmented, selecting meaningful pairs to query humans is yet another challenge: apples-to-apples comparisons are more meaningful.

We propose a simple solution to these challenges: scene graph–based trajectory segmentation and selection. The abstract state dimensions in the scene graphs naturally overlap with the reward features, since these scene graphs are designed to contain information that is critical to task success. The key observation is that in long-horizon tasks, a trajectory consists of multiple abstract state transitions, and *two consecutive abstract state transitions* naturally define the starting and ending points of a segment, i.e., a segment corresponds to only *one* abstract state $g$. Then, for query selection, we select a pair of segments with two abstract states $g$ and $g'$, such that $d(g, g') = 1$ where $d$ denotes the hamming distance (recall that $g$ is a binary vector). In other words, $g$ and $g'$ are two abstract states that only differ in one dimension. Hence the remaining dimensions are held constant, leading to a controlled comparison. If we cannot find a pair of segments such that $d(g, g') = 1$, we choose the pair that has the minimum hamming distance. The algorithm, **DRPL**, or Dual Representation based Preference Learning, is shown in Algorithm 2. We use a standard reward weight estimation algorithm (a Bayesian inverse reinforcement learning algorithm) that is based on the low-level representation [41, 78], the only difference is how we segment and select trajectories for a query.

## 4 Experiments

We test our algorithms in five continuous control tasks (Fig. 2) for both evaluative feedback and preference learning. The following describes the dual representation (low and high level) of each task.

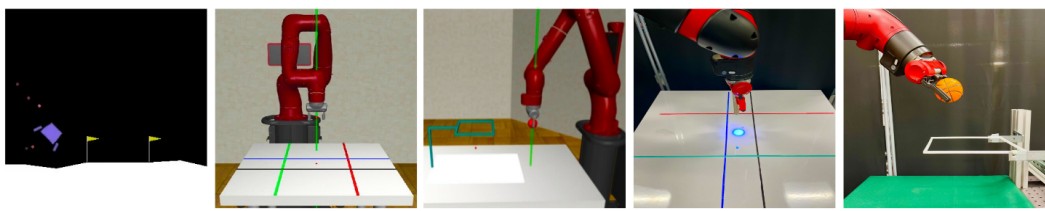

Figure 2: Five continuous control tasks in our experiments: `Lunar-Lander`, `Reaching-Sim`, `Placing-Sim`, `Reaching-Real`, and `Placing-Real`.

**Low-level representation.** In `Lunar-Lander-Continuous-v2` [79], we control a spaceship to land in the middle of two flags without crashing. The state, $s \in \mathbb{R}^8$, includes position, velocity, and leg-contact information, and the action signals, $a \in \mathbb{R}^2$ control the engines. In `Reaching-Sim` and `Reaching-Real`, the robot's goal is to move its gripper, to the center area marked on the table from a random starting location. The state, $s \in \mathbb{R}^2$, specifies the position of the gripper in the $xy$-plane and the actions, $a \in \mathbb{R}^2$, are delta movements on that plane. In `Placing-Sim` and `Placing-Real`, the robot must move an initially grasped ball from a random location into a hoop and drop it there. The state, $s \in \mathbb{R}^4$, includes the gripper position in 3D space and the gripper state (close/open). The actions, $a \in \mathbb{R}^4$, are delta movements in 3D Cartesian space and the gripper control. The simulation tasks are implemented in the Robosuite [80]. Further details can be found in Appendix 1.

**High-level representation.** Fig. 1 depicts the symbolic scene graph for `Placing-` tasks, where nodes correspond to task-relevant objects, and edges encode binary relations between them. Appendix 1 includes further details about the tasks and their symbolic scene graphs.

### 4.1 Results: Evaluative Feedback

We use synthetic humans in simulated environments and real humans in real-world environments. For synthetic humans (a fully trained SAC agent, more details in Appendix 2), the learning agent chooses an action $a$ in state $s$, and the oracle chooses an action $a^*$. The oracle SAC computes the Q values for these actions: $Q(s,a)$ and $Q(s,a^*)$. If the learning agent chooses an action that has a Q-value close enough to $Q(s,a^*)$, it is a good action and the agent should receive positive feedback. Otherwise, it should receive negative feedback: $H(s,a) = +1$, if $Q(s,a) \geq \alpha Q(s,a^*); -1$ otherwise. The $\alpha$ increases over time (see Appendix 2) to encourage the agent to learn to choose better actions during training.

We compare our method to the following baselines in simulated environments: (a) the SAC baseline without any human feedback (b) EF-100%, EF-50%, and EF-25%: the agent asks for feedback with a probability of 100%, 50%, or 25% at every timestep. Hyperparameters of all the algorithms can be found in Appendix 2.

Results for simulation (averaged across 5 seeds) are shown in Fig. 3. Our algorithm DREF (shown in blue) achieves comparable performance to EF-100% with only 6.2%, 1.6%, and 8.4% feedback in `Lunar-Lander`, `Reaching-Sim`, and `Placing-Sim`, respectively, corresponding to approximately $16\times$, $62\times$, and $11\times$ improvement in feedback sample efficiency.

For real robot experiments with six humans per task, we omit the EF-100% baseline since it requires step-by-step feedback from humans for a very long period of time. Humans are also allowed to choose "no feedback" if they prefer. The order of the three methods is randomized to counterbalance the ordering effect (see Appendix 4). Fig. 4 shows the results. In `Reaching-Real`, DREF achieves comparable performance to EF-50% with 13.8% feedback, leading to a $3\times$ improvement in human feedback sample efficiency. `Placing-Real` is extremely challenging due to the long task horizon and sparsity in reward signals. DREF achieves a better performance than all baselines, by a large margin, with only 17% human feedback.

The human experiments are followed by a survey (details in Appendix 4.1) asking humans about their overall experience training the robots (E), perceived intelligence level (I), and cognitive ease (C) (see Fig. 4). It is evident that our algorithm leads to a better user experience and reported effort (additional analyses are in Appendix 4.2). These results are supported by the observation that the total training time and no feedback count are lower for DREF. This may also explain why EF-50% performs poorly in the challenging `Placing-Real` task: the training is laborious and cognitively demanding with this amount of guidance, and humans are prone to errors in this process.

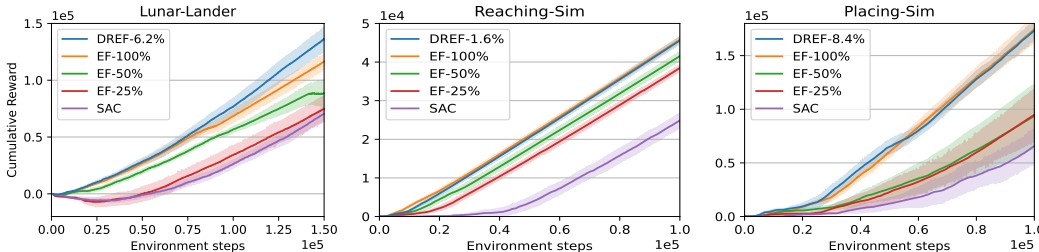

Figure 3: Cumulative rewards gained during training for `Lunar-Lander`, `Reaching-Sim`, and `Placing-Sim`. The percentage corresponds to the percentage of feedback provided by the oracle during training. The proposed algorithm, DREF, achieves comparable performance with EF-100% with much less feedback and outperforms all other baselines. Error bars indicate the standard error of the means ($n = 5$).

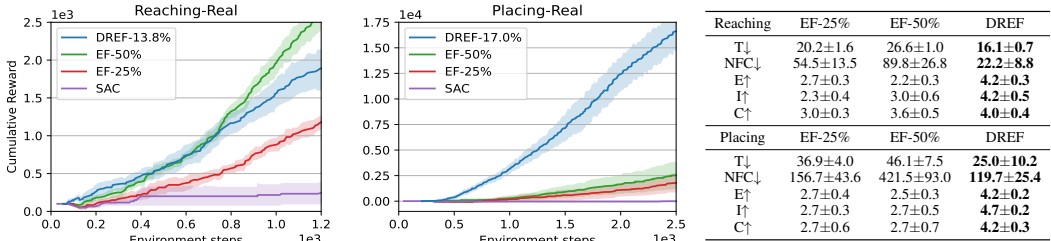

Figure 4: Cumulative rewards gained during training for `Reaching-Real`, and `Placing-Real`. The proposed algorithm, DREF, achieves comparable or better performance with 50% feedback with much less feedback and outperforms all other baselines. Error bars indicate the standard error of the means ($n = 6$). Post-completion user survey (5-point Likert scale) results indicate that DREF leads to better user experience, perceived intelligence, and less reported effort ($n = 6$). T: total training time, NFC: number of "no feedback" responses, E: overall experience, I: perceived intelligence, C: cognitive ease. See Appendix 4.1 and 4.2 for survey design and additional analyses.

To conclude, the results strongly support our hypothesis about evaluative feedback: asking for human feedback sensibly leads to a better learning outcome. DREF achieves this by implementing uncertainty-aware active learning within the dual representation framework.

## 4.2 Results: Preference Learning

We now present the results of preference learning experiments. Similar to evaluative feedback, we use synthetic humans in simulated environments and real humans in real-world environments. The synthetic human has access to the true reward weight $\theta$ (more details in Appendix 3). For every pair of queries$(\xi, \xi')$, the oracle calculates the true reward $r$ and $r'$. The oracle returns $q(\xi, \xi') = 0$ if $r(\xi) > r(\xi')$, and $q(\xi, \xi') = 1$ otherwise. The trajectories are generated by agents starting at random positions and taking random actions. We record such trajectories, store them, and select trajectories or segments to query synthetic or real humans. Preference learning algorithms update the posterior belief of the reward function after each query. We use cosine similarity as the alignment score to measure the distance between the estimated reward weights $\hat{\theta}$ and true weights $\theta$ [35, 41].

We compare DRPL with three baselines: (a) full trajectory query: randomly select two full trajectories to query; (b) random fragment query: randomly select two full trajectories, and cut one fragment out of each with a length equal to the average length of DRPL queries; (c) DRPL-SS: a version of our algorithm that selects a pair of trajectories associated with the *same* abstract state, instead of two abstract states that differ in one dimension. Further details can be found in Appendix 3.

Results for simulation (averaged across 5 seeds with 5 different true rewards) are shown in Fig. 5. DRPL (blue) converges to an alignment score around or greater than 0.8 for all tasks and outperforms all other algorithms. DRPL-SS performs better than the other baselines but is worse than DRPL, indicating that both components of our DRPL (segmentation and selection) are important.

Results for real robot experiments with humans (six for each task) are shown in Fig. 6. We omit the full trajectory baseline due to the time required to compare long trajectories generated by the random

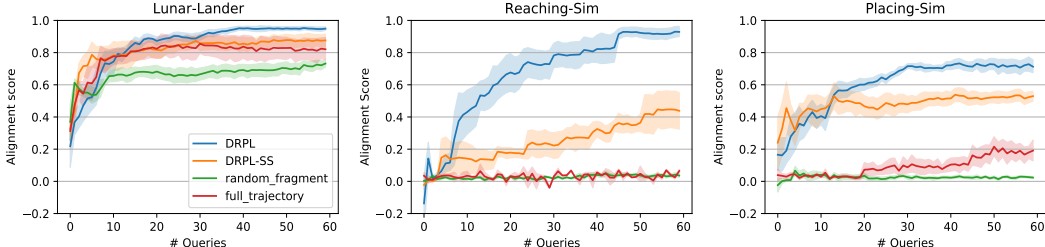

Figure 5: Alignment scores [35, 41] for `Lunar-Lander`, `Reaching-Sim`, and `Placing-Sim`. DRPL performs the best upon convergence. Error bars indicate the standard error of the means ($n = 5$).

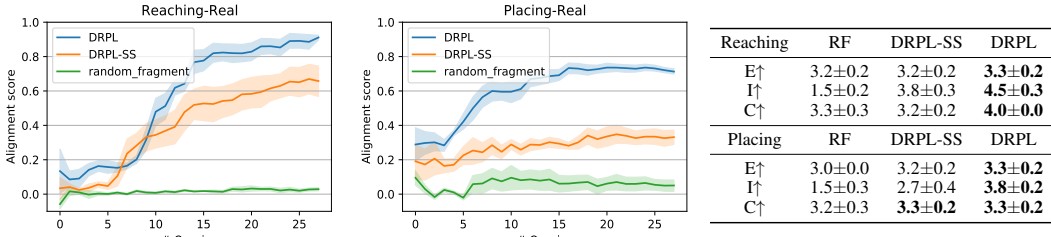

| Reaching | RF | DRPL-SS | DRPL |
|---|---|---|---|
| E↑ | 3.2±0.2 | 3.2±0.2 | **3.3±0.2** |
| I↑ | 1.5±0.2 | 3.8±0.3 | **4.5±0.3** |
| C↑ | 3.3±0.3 | 3.2±0.2 | **4.0±0.0** |
| Placing | RF | DRPL-SS | DRPL |
| E↑ | 3.0±0.0 | 3.2±0.2 | **3.3±0.2** |
| I↑ | 1.5±0.3 | 2.7±0.4 | **3.8±0.2** |
| C↑ | 3.2±0.3 | **3.3±0.2** | **3.3±0.2** |

Figure 6: Alignment scores [35, 41] for `Reaching-Real` and `Placing-Real`. DRPL achieves the best performance upon convergence. Error bars indicate the standard error of the means ($n = 6$). Post-completion survey (5-point Likert scale) results indicate that DRPL leads to better perceived intelligence. E: overall experience, I: perceived intelligence, C: cognitive ease. See Appendix 4.3 and 4.4 for survey design and additional analyses.

agents. Our algorithm DRPL (shown in blue) achieves better performance than both baselines. Similar to evaluative feedback, the user experience survey (Fig. 6) shows that DRPL leads to better perceived intelligence (see Appendix 4.3 and 4.4). To conclude, the results support our key insight about preference learning: abstract state–based trajectory segmentation and query lead to a better learning outcome. DRPL is an instantiation of this idea within the dual representation framework.

## 5  Discussion

To summarize, we have proposed a dual representation framework for robot learning from human guidance. In the context of robot learning, abstract representation has long been a research topic. However, adopting it in decision learning often comes with a price: the critical information needed to learn a good policy may be lost in the process of abstraction. Researchers typically avoid this problem by having a hierarchical representation. The key difference is that we only use abstract representation as an auxiliary representation.

We show that the abstract scene graph representation allows us to utilize important heuristics that facilitate training in two popular forms of human guidance: evaluation and preference, both of which utilize human evaluations for observed agent policies. Evaluative feedback targets state-action pairs that are fine-grained, while preference learning targets trajectories that could be too coarse. Hence the former needs a grouping mechanism for generalization, and the latter needs a segmentation mechanism for efficient queries. Our proposed framework is a unified approach to provide both.

We demonstrate the effectiveness of our approaches in five challenging continuous control tasks. Our algorithms show significant improvements in performance and reduction in human effort, compared to algorithms without an auxiliary scene graph representation. These improvements make learning from human guidance methods significantly more appealing for real-robot learning.

**Limitations.** Currently, our proposed approaches leverage expert-defined scene graphs. Although this representation should be closer to the human representation, the actual representations are likely to be different across human trainers. Adapting to different abstract representations may further improve learning outcomes and user experience.

**Acknowledgement**

The work is in part supported by the Stanford Institute for Human-Centered AI (HAI), ONR MURI N00014-22-1-2740, ONR MURI N00014-21-1-2801, Amazon, Analog, IBM, JPMC, Meta, Salesforce, and Samsung. Ruohan Zhang is supported by Wu Tsai Human Performance Alliance Fellowship.

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
