# OpenReview forum: "A Dual Representation Framework for Robot Learning with Human Guidance"
_robot-learning.org/CoRL/2022/Conference — CoRL 2022 Poster_

### Official Review · Reviewer_JYfb · 2022-07-28

**Originality:** Good
**Technical Quality:** Very Good
**Clarity Of Presentation:** Very Good
**Impact:** 3

**Recommendation:**

Weak Accept: I recommend accepting the paper, but will not argue for my recommendation if the majority of other reviewers have a different opinion.

**Summary:**

The authors present an approach for learning robot skills under human guidance. They propose two independent algorithms, one for learning from human evaluative feedback and one for learning from human preferences. The basis for their approach are two different representations, one continuous representation (states and actions) of the robot and one discrete representation of the task. While the continuous representation is used for training the RL agent, the discrete more abstract representation of the feedback is used for deciding when and how to ask the human for guidance.

**Issues:**

- From reading the paper, it is unclear how the scalar guidance signal H is genereated for the synthetic human experiments. I had to read the Appendix in order to understand how it is done. In my opinion the explanation must be in the paper and not in the Appenxid since it is crucial for understanding the learning process.

- Line 126 receveid -> received

- I was confused by the term "alignment score". Is it an established term in the preference learning / human guidance community? I guess you chose the term since it measure how well the predicted preference aligns with the true human preference. From a learning perspective, in your context, it is the error between the predicted and actual reward and therefore I would simply use a more general term like "Avg. Prediction Error". If you think alignment score is more fitting, I would recommend adding some explanation to the text (and maybe to Figure 5).

**Quality Of The Limitations Section:**

Limitations are addressed clearly

**Reviewer Expertise:**

3: The reviewer is fairly confident that the evaluation is correct

**Robotics Focus:**

Sufficient demonstration on hardware

**Strengths And Weaknesses:**

Strengths
- The paper is well written and straightforward to follow
- Number of experiments is adequate (both simulation and real experiments)

Weaknesses
- According to my understanding the algorithms are not mutually exclusive. I do not understand why there is no experiment where it is tested if using both algorithms at the same time further improves the learning efficiency. If the algorithms can't be combined, the authors should clarify why.
- Scientifically, it would have been interesting to evaluate how different scene graph representations of the same task effect the learning efficiency and task performance. However, I also think this could be follow-up work that does not have to be included in this paper.
- The tasks in the experiments can be solved by an unambiguous state-action pair mapping (no history of actions is required for solving a task). Therefore, the RL agent does not need to know the scene graph which otherwise would be needed to disambiguate the continuous states. Therefore, no coupling between the two representations is necessary and they can be kept independent.

**Summary Of Recommendation:**

Overall, I think it is a solid paper without bigger weaknesses. With the dual representation, the authors shifted a lot of the complexity of the learning problem to the designer of the scene graph. Designing a good scene graph is crucial for task success, as the authors also stated in the limitations section. Therefore, I see this paper as an interesting first step in the right direction.

---

> ### Author Response · Authors · 2022-08-26
> **Response to Reviewer JYfb**
>
> Thank you for the constructive comments and helpful feedback.
>
> **RJYfb-Q1: According to my understanding the algorithms are not mutually exclusive. I do not understand why there is no experiment where it is tested if using both algorithms at the same time further improves the learning efficiency. If the algorithms can't be combined, the authors should clarify why.**
>
> RJYfb-A1: The two algorithms are not directly combinable. Evaluative feedback is an RL problem, in which we train an RL agent to maximize reward gained from the environment. Preference learning is an IRL problem (without interacting with the environment), we try to estimate the unobserved reward function from paired queries. To the best of our knowledge, these two learning paradigms are fairly independent. Although from a representation perspective, there is a connection between them as shown by this work.
>
> **RJYfb-Q2: Scientifically, it would have been interesting to evaluate how different scene graph representations of the same task effect the learning efficiency and task performance. However, I also think this could be follow-up work that does not have to be included in this paper.**
>
> RJYfb-A2: Thanks for suggesting the follow-up work. We agree that this is very interesting and will explore it further.
>
> **RJYfb-Q3: The tasks in the experiments can be solved by an unambiguous state-action pair mapping (no history of actions is required for solving a task). Therefore, the RL agent does not need to know the scene graph which otherwise would be needed to disambiguate the continuous states. Therefore, no coupling between the two representations is necessary and they can be kept independent.**
>
> RJYfb-A3: Partial observability does not limit our proposed framework as long as the agent has enough information to tell which abstract state it is in. In that case, we need to change the RL agent to be an RL+memory agent, it will still keep the scene graph representation for the proposed algorithms.
>
> **RJYfb-Q4: it is unclear how the scalar guidance signal H is genereated for the synthetic human experiments. I had to read the Appendix in order to understand how it is done. In my opinion the explanation must be in the paper and not in the Appenxid since it is crucial for understanding the learning process.**
>
> RJYfb-A4: We have added that information in the manuscript (revision, Line 206 - 212).
>
> **RJYfb-Q5: Line 126 receveid -> received**
>
> RJYfb-A5: Corrected.
>
> **RJYfb-Q6: I was confused by the term "alignment score". Is it an established term in the preference learning / human guidance community? I guess you chose the term since it measure how well the predicted preference aligns with the true human preference. From a learning perspective, in your context, it is the error between the predicted and actual reward and therefore I would simply use a more general term like "Avg. Prediction Error". If you think alignment score is more fitting, I would recommend adding some explanation to the text (and maybe to Figure 5).**
>
> RJYfb-A6: Alignment score is a metric used in preference learning (Sadigh et al., 2017; Bıyık & Sadigh, 2018), which is the cosine similarity distance between the estimated reward weights and actual weight. We use the Bayesian preference learning algorithm from these previous works so we choose to use their evaluation metrics as well. We have clarified these in the text (revision, Line 248).

---

### Official Review · Reviewer_8h9i · 2022-07-28

**Originality:** Good
**Technical Quality:** Fair
**Clarity Of Presentation:** Good
**Impact:** 3

**Recommendation:**

Weak Accept: I recommend accepting the paper, but will not argue for my recommendation if the majority of other reviewers have a different opinion.

**Summary:**

This work advocates for aligning robot representations with human internal state representations. The core hypothesis of this work is that a symbolic scene graph (where objects are nodes and relationships between them are edges) is an effective representation for such alignment. With this idea, the paper proposes two learning algorithms: learning from evaluative human feedback (which is formulated as a multi-armed bandit) and learning from human preferences. The idea and algorithms are evaluated with simulated and real human data and on simulated and real robots.


**Issues:**

The following are summarized from the weaknesses mentioned above:

–Please elaborate on the disconnect between the paper’s idea / claim and the proposed evaluation. The hypothesis that scene graphs are a useful representation is very interesting, but I would expect much of the evaluations to be focused on comparing different existing representations to the proposed one. For example, one could compare the scene-graph representation vs. assuming that the feature space used in the reward function is the person’s representation vs. extracting some learned latent state representation in an unsupervised way (e.g., via autoencoders) vs. having no explicit intermediate representation. I believe further evaluation is needed to properly test the core hypothesis of this paper: that scene-graph representations help humans teach robots more effectively as compared to alternatives (like no intermediate representation or other representations).

–The paper proposes two alternative learning methods for the evaluation-based and preference-based learning settings. Please clarify the connection between them and the need for the scene-graph itself. For example, in preference-based learning, the robot has a reward represented as a linear combination of features. However, this reward is not used in the evaluation-based learning setting – why not? Furthermore, what is the relationship between the reward function and the scene graph representation? It seems like the scene graph could be used as a “featurization”, and then existing SOTA preference-based learning methods (like the ones cited in the related work) could be leveraged. Why is it desirable to decouple the graph-based representation from the reward featurization in the proposed method? Answering these questions both through the text and through ablation studies would significantly strengthen the paper’s claim.



**Quality Of The Limitations Section:**

Limitations are addressed clearly

**Reviewer Expertise:**

4: The reviewer is confident but not absolutely certain that the evaluation is correct

**Robotics Focus:**

Sufficient demonstration on hardware

**Strengths And Weaknesses:**

Strengths:
--------------
–Interesting and timely choice of problem that is valuable to the HRI community

–Interesting proposed representation of a scene graph

–Experiments in both simulation and with a real robot, and with simulated and real people

Weaknesses:
-----------
–There seems to be a disconnect between the paper’s idea / claim and the proposed evaluation. The hypothesis that scene graphs are a useful representation is very interesting, but I would expect much of the evaluations to be focused on comparing different existing representations to the proposed one. For example, one could compare the scene-graph representation vs. assuming that the feature space used in the reward function is the person’s representation vs. extracting some learned latent state representation in an unsupervised way (e.g., via autoencoders) vs. having no explicit intermediate representation. However, the comparisons focus only on different methods for querying for human feedback (e.g., SAC model with no human feedback vs. requesting for human feedback 50% of the time, etc.). While a very important comparison, I think this is not properly testing the core hypothesis of this paper: that scene-graph representations help humans teach robots more effectively as compared to alternatives (like no intermediate representation or other representations).

–The paper proposes two alternative learning methods for the evaluation-based and preference-based learning settings. While it is ok to have different algorithmic approaches for different types of feedback, it was difficult to understand their connection or their need for the scene-graph itself. For example, in preference-based learning, the robot has a reward represented as a linear combination of features. However, this reward is not used in the evaluation-based learning setting – why not? Furthermore, what is the relationship between the reward function and the scene graph representation? It seems like the scene graph could be used as a “featurization”, and then existing SOTA preference-based learning methods (like the ones cited in the related work) could be leveraged. Why is it desirable to decouple the graph-based representation from the reward featurization in the proposed method? Answering these questions both through the text and through ablation studies would significantly strengthen the paper’s claim.

Section 2:

–”We explore how a symbolic scene graph could also be a useful intermediate representation for human-in-the loop robot learning, under the assumption that it aligns better with human representation.” What is the symbolic scene graph being compared to here? It would be helpful to explicitly mention here that the symbolic scene graph is hypothesized to do better than <insert the other approaches here> (e.g., learning without intermediate representations, learned latent-state representations, hand-crafted feature representations, etc.)

–The related work could be strengthened by discussing the representation learning literature from machine learning, and the similarities / differences from the proposed work. For example:

[1] X. Chen, Y. Duan, R. Houthooft, J. Schulman, I. Sutskever, and P. Abbeel, “Infogan: Interpretable representation learning by information maximizing generative adversarial nets,” in Proceedings of the 30th International Conference on Neural Information Processing Systems,
ser. NIPS’16.

[2] I. Higgins, L. Matthey, A. Pal, C. P. Burgess, X. Glorot, M. M. Botvinick, S. Mohamed, and A. Lerchner, “beta-vae: Learning basic visual concepts with a constrained variational framework,” in ICLR, 2017.

[3] R. T. Q. Chen, X. Li, R. B. Grosse, and D. K. Duvenaud, “Isolating sources of disentanglement in variational autoencoders,” in Advances in Neural Information Processing Systems, S. Bengio, H. Wallach, H. Larochelle, K. Grauman, N. Cesa-Bianchi, and R. Garnett, Eds., vol. 31. Curran Associates, Inc., 2018.

[4] D. Brown, R. Coleman, R. Srinivasan, and S. Niekum, “Safe imitation learning via fast Bayesian reward inference from preferences,” in Proceedings of the 37th International Conference on Machine Learning, 2020.

–Some related work that would be valuable to include (the former is about aligning robot representations with human representations, the latter is work on online learning from human evaluative feedback):

[1] Bobu, Andreea, et al. "Inducing structure in reward learning by learning features." The International Journal of Robotics Research (2022): 02783649221078031.

[2] Jain, Ashesh, et al. "Learning preferences for manipulation tasks from online coactive feedback." The International Journal of Robotics Research 34.10 (2015): 1296-1313.

Section 3:

–The claim ”Integrating these methods with continuous control enables efficient uncertainty-aware active learning from humans” should be substantiated a bit more in the text. For example, are there prior works that achieve this? If so, please cite them.

–”The agent learns a policy to maximize human feedback and environment reward simultaneously.” This objective seem counter-intuitive – it should be desirable to find a policy which *minimizes* human feedback (to decrease human effort) and *maximizes* environmental reward? Perhaps this was a typo.

–”In this way, the dual representation leads to efficient uncertainty-aware active learning from human feedback.” This claim should also be substantiated with a proof, numerical results, or a citation, since the presentation of the algorithm does not immediately demonstrate that the proposed dual representation yields more efficient active learning.

Section 4:

–The right-most table in Figure 4 is difficult to decipher. Are these the results of the Likert Scale Survey? If so, it would be helpful to title it as such. Additionally, it would be helpful to include the specific questions that were asked in the survey, as well results of an ANOVA and the p-values for each question to indicate if the results are statistically significant.



**Summary Of Recommendation:**

While the problem choice and high-level idea (of using a scene graph as a way to align human representations with robot representations) are relevant and interesting, I am concerned that there is a disconnect between the proposed approach and the way in which it has been evaluated. Specifically, I believe that more experiments need to be conducted to compare the scene-graph representation to alternative representations, and show how the proposed method hits the sweet spot of low-sample complexity / low perceived human effort while achieving high reward when compared to alternative representations (e.g., hand-engineered featurizations, automatically-extracted featurizations, etc.). Thus, I recommend weak reject.

==================================================
After discussion, I changed my recommendation to "weak accept".

---

> ### Author Response · Authors · 2022-08-26
> **Response to Reviewer 8h9i 1/2**
>
> Thank you for the constructive comments and helpful feedback.
>
> **R8h9i-Q1: There seems to be a disconnect between the paper’s idea / claim and the proposed evaluation. … but I would expect much of the evaluations to be focused on comparing different existing representations to the proposed one. For example, one could compare the scene-graph representation vs. assuming that the feature space used in the reward function is the person’s representation vs. extracting some learned latent state representation in an unsupervised way (e.g., via autoencoders) vs. having no explicit intermediate representation. However, the comparisons focus only on different methods for querying for human feedback (e.g., SAC model with no human feedback vs. requesting for human feedback 50% of the time, etc.). While a very important comparison, I think this is not properly testing the core hypothesis of this paper…**
>
> R8h9i-A1: Learned representation is definitely an interesting direction to explore. However, we want to emphasize the following points:
> - Not all forms of learned representation would work. First, without supervision, it may be difficult to learn a representation that matches human representation. Second, the learned representation must be symbolic / discrete to enable the proposed algorithms.
> - The focus of this work is to make evaluative feedback (EF) and preference learning (PL) more efficient for physical robots trained with real humans. With EF, we trained deep RL agents to perform two challenging tasks in only 1,250/2,500 steps. Many representation learning methods are infeasible with this amount of data. For example, we experimented with the information bottleneck-based method proposed by [R1] to learn abstract representations (replacing imitation learning with evaluative feedback), and we were not able to achieve any meaningful results in 2 million steps in simulation. Note the original work uses demonstrations which is a stronger supervision signal than EF, and they require 2000 episodes to train on a simple game Breakout. Hence, we chose to use human-defined representation which is more sample efficient.
> - “Feature space used in the reward function is the person’s representation” ->  The reward features overlap with abstract state dimensions in the scene graphs, since these scene graphs are designed to contain information that is critical to task success. We have clarified this in the revised manuscript (Line 174).
> - Finally, we understand your concern about the mismatch between our claim and evaluation methods. We compare scene graph representation with no additional representation, and demonstrate that this auxiliary representation enables more efficient querying methods. We understand that our presentation causes confusion, and have revised our manuscript to make our contributions more clearly defined (Line 283).
> [R1] Abel, D., et al., (2019). State abstraction as compression in apprenticeship learning. AAAI.
>
> **R8h9i-Q2: The paper proposes two alternative learning methods for the evaluation-based and preference-based learning settings. While it is ok to have different algorithmic approaches for different types of feedback, it was difficult to understand their connection or their need for the scene-graph itself.**
>
> R8h9i-A2: Our goal is to show the proposed framework is useful for more than one popular human-in-the-loop robot learning paradigm. Additionally, there exists a connection between the two methods. Evaluative feedback (EF) and preference learning (PL) both elicit human evaluations for observed agent policies. EF targets state-action pairs which are fine-grained, while most PL targets full trajectories which are too coarse. Hence the former needs a grouping mechanism (for generalization, e.g., to estimate the feedback prediction error as in Eq. 2), and the latter needs a segmentation mechanism. Using abstract states is a unified approach to provide both. We have added this discussion to the revised manuscript in related work (revision, Line 72 and Line 80) and discussion (Line 277-280).
>
> **R8h9i-Q3: …For example, in preference-based learning, the robot has a reward represented as a linear combination of features. However, this reward is not used in the evaluation-based learning setting – why not?**
>
> R8h9i-A3: The rich set of reward features is used in preference learning (an IRL problem) to better capture the diversity in human preference. e.g., although the Placing tasks only require the ball to be put into the hoop, some humans may prefer the ball to be dropped from a lower altitude.  In evaluative feedback (an RL problem), the goal is to maximize the environment reward so these features are unnecessary.

---

> > ### Author Response · Authors · 2022-08-26
> > **Response to Reviewer 8h9i 2/2**
> >
> > **R8h9i-Q4: Furthermore, what is the relationship between the reward function and the scene graph representation?...Why is it desirable to decouple the graph-based representation from the reward featurization in the proposed method?**
> >
> > R8h9i-A4: They are not decoupled. The reward features naturally overlap with abstract state dimensions in the scene graphs, since these scene graphs are designed to contain information that is critical to task success. We have clarified this in the revised version (Line 174).
> >
> > **R8h9i-Q5: ”We explore how a symbolic scene graph could also be a useful intermediate representation for human-in-the loop robot learning, under the assumption that it aligns better with human representation.” What is the symbolic scene graph being compared to here? It would be helpful to explicitly mention here that the symbolic scene graph is hypothesized to do better than <insert the other approaches here> (e.g., learning without intermediate representations, learned latent-state representations, hand-crafted feature representations, etc.)**
> >
> > R8h9i-A5: Here we hypothesize that the symbolic scene graph aligns better with human representation than the low-level, raw state representation. But please see our response to RM5fV-Q3, in which we have removed all claims related to human internal representations.
> > We have revised the writing here (revision, Line 94-97) to be: We explore how a symbolic scene graph could be a useful additional representation for human-in-the-loop robot learning. We hypothesize that it allows the learning agent to query humans more efficiently and learn with less human guidance than using the low-level, raw state representation alone.
> >
> > **R8h9i-Q6: The related work could be strengthened by discussing the representation learning literature from machine learning, and the similarities / differences from the proposed work. Some related work that would be valuable to include (the former is about aligning robot representations with human representations, the latter is work on online learning from human evaluative feedback).**
> >
> > R8h9i-A6: Thanks for these references. We have included all of them in the related work section. Although based on our understanding, Jain et al., (2015) belongs to the preference learning paradigm.
> >
> > **R8h9i-Q7: The claim ”Integrating these methods with continuous control enables efficient uncertainty-aware active learning from humans” should be substantiated a bit more in the text. For example, are there prior works that achieve this? If so, please cite them.**
> >
> > R8h9i-A7: Here we are specifically referring to the Upper Confidence Bound (UCB) method.  To the best of our knowledge, active learning with UCB in robot learning with humans has not been achieved before. To be more concrete, we have modified the sentence to be: “This formulation brings the opportunities to use formal methods designed for discrete state and action spaces, such as Upper Confidence Bound (UCB1). Integrating UCB1 with continuous control enables efficient uncertainty-aware active learning from humans.” (revision, Line 124-126)
> >
> > **R8h9i-Q8: The agent learns a policy to maximize human feedback and environment reward simultaneously.” This objective seems counter-intuitive – it should be desirable to find a policy which minimizes human feedback (to decrease human effort) and maximizes environmental reward? Perhaps this was a typo.**
> >
> > R8h9i-A8: By “maximize human feedback” we mean “maximize expected positive feedback from humans.” The text is updated to clarify this (revision, Line 146).
> >
> > **R8h9i-Q9: ”In this way, the dual representation leads to efficient uncertainty-aware active learning from human feedback.” This claim should also be substantiated with proof, numerical results, or a citation, since the presentation of the algorithm does not immediately demonstrate that the proposed dual representation yields more efficient active learning.**
> >
> > R8h9i-A9: You are right: this is the hypothesis we want to test. We have changed this to be: We hypothesize that in this way, the dual representation could lead to efficient uncertainty-aware active learning from human feedback, which we will demonstrate in our experiments (revision, Line 156).
> >
> > **R8h9i-Q10: The right-most table in Figure 4 is difficult to decipher. Are these the results of the Likert Scale Survey? If so, it would be helpful to title it as such. Additionally, it would be helpful to include the specific questions that were asked in the survey, as well results of an ANOVA and the p-values for each question to indicate if the results are statistically significant.**
> >
> > R8h9i-A10: Please see Appendix 4 for the survey questions. We have included new results on repeated measures ANOVA tests in revised Appendix 4.2 and 4.4. Overall, the results support the original claim in the paper.

---

### Official Review · Reviewer_32aJ · 2022-07-31

**Originality:** Good
**Technical Quality:** Good
**Clarity Of Presentation:** Good
**Impact:** 3

**Recommendation:**

Weak Accept: I recommend accepting the paper, but will not argue for my recommendation if the majority of other reviewers have a different opinion.

**Summary:**

The paper presents methods for learning from human guidance that leverage scene graphs to more efficiently elicit feedback. In the evaluative feedback setting, a UCB-inspired algorithm is developed to query humans based on prediction error in the preferences for given abstract states. In the learning from preferences setting, the proposed algorithm selects query trajectory pairs based on similar abstract states. It is shown that this scheme yields better asymptotic performance in inferring a reward function. On the whole, the paper demonstrates that formulating queries based on abstract states rather than low-level states is an effective mechanism for obtaining quality feedback from human users.

**Issues:**

A rather similar idea was explored in the paper [Active Task-Inference-Guided Deep Inverse Reinforcement Learning](https://arxiv.org/pdf/2001.09227.pdf), though I think the novelty claim of this paper is preserved. It’s worth a read through [Memarian’s thesis](https://repositories.lib.utexas.edu/handle/2152/88568) nonetheless.
Consider citing and/or using recent work from Scott Niekum’s group on learning from preferences - see “A Ranking Game for Imitation Learning”.

I’m a bit perplexed surrounding the incorporation of the human feedback into the learning objective for the ER setting. It seems the model is trained with equation (3) to predict the scalar feedback signal $H$, not necessarily to maximize it. It is stated that the agent is trained to maximize human feedback and environment reward simultaneously, but the given loss function seems to minimize the standard SAC loss plus the difference between $\hat H$ and $H$, minimizing the prediction error about the human feedback (rather than maximizing it’s value). TAMER+RL, DQN-TAMER etc. suggest several ways in which to incorporate the feedback into the RL objective, e.g.  incorporating into the value targets, or choosing actions from some combination of Q and H, but these do not appear to be in use here. I suppose the gradients of the feedback prediction term are providing some learning signal?

typo line 126 “receveid”

**Quality Of The Limitations Section:**

Limitations are addressed clearly

**Reviewer Expertise:**

3: The reviewer is fairly confident that the evaluation is correct

**Robotics Focus:**

Sufficient demonstration on hardware

**Strengths And Weaknesses:**

The paper is well-motivated and studies an important problem. The proposed algorithms demonstrate clear improvements in performance over the baselines. The paper is generally well-written and clear.

The learning-from-preferences experiment is a bit lacking. The data generation scheme (random trajectories) could be more efficient (e.g. something online, like DAgger, or using human demonstrations), and it would be nice to see learning curves using the estimated reward functions rather than just the alignment scores.

**Summary Of Recommendation:**

I’m recommending weak acceptance - while the evaluation has some issues, I think the core idea is novel and the results demonstrate the effectiveness of the method.

---

> ### Author Response · Authors · 2022-08-26
> **Response to Reviewer 32aJ**
>
> Thank you for the constructive comments and helpful feedback.
>
> **R32aJ-Q1: The learning-from-preferences experiment. The data generation scheme (random trajectories) could be more efficient (e.g. something online, like DAgger, or using human demonstrations), and it would be nice to see learning curves using the estimated reward functions rather than just the alignment scores.**
>
> R32aJ-A1: Thanks for the suggestion. We use a random policy because it provides good state coverage. We have actually experimented with near-optimal trajectories from demonstrations, but found that they did not provide enough coverage, so learning the true reward was difficult (e.g., crashing in Lunar Lander results in a penalty).
> Based on your suggestion, we have finished the learning curve for Luna-Lander (average of 5 runs for each algorithm). The recovered reward by DRPL leads to better learning outcomes compared to other algorithms. Please see attached file for the results. We will include the results in the Appendix once we finish experiments for all the tasks.
>
> **R32aJ-Q2: A rather similar idea was explored in the paper Active Task-Inference-Guided Deep Inverse Reinforcement Learning, though I think the novelty claim of this paper is preserved. It’s worth a read through Memarian’s thesis nonetheless. Consider citing and/or using recent work from Scott Niekum’s group on learning from preferences - see “A Ranking Game for Imitation Learning”.**
>
> R32aJ-A2: Thanks for these references. We have included both of them in the related work.
>
> **R32aJ-Q3: I’m a bit perplexed surrounding the incorporation of the human feedback into the learning objective for the ER setting…I suppose the gradients of the feedback prediction term are providing some learning signal?**
>
> R32aJ-A3: You are right. Due to the continuous action space, DQN-TAMER is not directly applicable. Hence we follow the standard actor-critic framework, in which human feedback is treated as another Q-value function. The critics (human and environment) estimate the value functions. The actor updates the policy distribution in the direction suggested by the weighted average of both critics. We have updated Eq. 3 (now Eq. 3-5) to separate the loss for the actor and losses for two critics to clarify this (revision, Line 141-143).

---

### Official Review · Reviewer_M5fV · 2022-08-05

**Originality:** Good
**Technical Quality:** Fair
**Clarity Of Presentation:** Very Good
**Impact:** 2

**Recommendation:**

Strong Reject: I recommend rejecting the paper and will argue for my recommendation even if other reviewers hold a different opinion.

**Summary:**

This paper proposed and evaluated two methods that reduce the number of times the robot asks for human feedback while learning different skills. The policies are learned in the continuous sensorimotor space. Yet, the robot exploits symbolic state information when it queries feedback/preferences from the humans. The main contribution of the paper is to exploit symbolic state information in deciding when and how to ask for human guidance/feedback. The first method enables the robot to ask for human feedback considering the robot's uncertainty about the symbolic states (based on their visitation count). The second method enables the robot to select a pair of trajectories again based on symbolic representations and ask which trajectory the humans prefer. The method is validated with 3 simulated and 2 real robot experiments. The method is mainly compared against a baseline that does not use any human feedback and also compared against baselines that use human feedback %N of the time.

**Issues:**

Some other comments:
- Symbolic scene graphs are emphasized. However, what I see is a binary vector that represents the environment. The full power of the graph structure is not used (e.g. object-object relations are not used). So emphasizing on the graph structure is not necessary.
- The claim on "humans adopting the slow system" in teaching is not well-justified. In teaching some skills, very low-level detailed trajectories might be important, in teaching some others, high-level representations.
- Lines 48-50: The correspondence between Evaluative Feedback and uncertainty awareness is not clear. The correspondence between Preference Learning and scene graph-based trajectory segmentation is not clear.
- Line 54: The authors missed important recent hybrid skill learning approaches such as Silver et al., 2022; Chitnes et al., 2021; Achterhold, 2021.
- The authors can include shared control reviews: Dragan and Srinivasa, 2012; Amirshirzad, 2019.
- Line 91: Unjustified statement. The tasks are probably represented at different levels also in humans.
- How "maximization of human feedback" is integrated into the loss in lines 131-132 is not clear.
- Line 136 (and line 7 in Algo 1): It is not clear exactly which state is queried from the ones ranked smaller than k.
- Line 141: Unjustified statement related to being "meaningful" to humans.
- Line 154: Related to the length of the trajectories: Why does your initial human insight idea not work here? No human would sum up any reward, they would simply say this is better than the other.
- Is IRL implemented in Section 3.3? What are p and P (also in Algo 2)?
- Line 209: "less effort": Effort was not measured. use "reported effort"
- What is alignment score? Where is it defined? Was it not possible to provide the results using a common metric in Sections 4.1 and 4.2?


Silver, Tom, et al. "Learning Neuro-Symbolic Skills for Bilevel Planning." arXiv preprint arXiv:2206.10680 (2022).
Chitnis, Rohan, et al. "Learning neuro-symbolic relational transition models for bilevel planning." arXiv preprint arXiv:2105.14074 (2021).
Achterhold, Jan, et al. "Learning Temporally Extended Skills in Continuous Domains as Symbolic Actions for Planning." arXiv preprint arXiv:2207.05018 (2022).
Dragan, Anca D., and Siddhartha S. Srinivasa. Formalizing assistive teleoperation. MIT Press, July, 2012.
Amirshirzad et al. "Human adaptation to human-robot shared control." IEEE Transactions on Human-Machine Systems 49.2 (2019): 126-136.



**Quality Of The Limitations Section:**

Additional details required

**Reviewer Expertise:**

4: The reviewer is confident but not absolutely certain that the evaluation is correct

**Robotics Focus:**

Sufficient demonstration on hardware

**Strengths And Weaknesses:**

- The paper is well-written and clear.
- The experimental evaluation is thorough in terms of the number and diversity of both simulated and real robot experiments.
- Linking and exploiting representations at different levels is important and have the potential to make a high impact in general. The robots need to learn low-level sensorimotor control, but for generalization and for high-level cognitive skills such as planning, reasoning, and communication, high-level symbolic knowledge becomes important. We need architectures that effectively combine different levels of representations. In this sense, the paper has the potential to make important contributions to the literature.
- I have three major comments:
1- The paper lacks providing a coherent and powerful message as it proposes, implements and evaluates two different human guidance methods with almost no connection in between. I can understand that the main point is that both of these human guidance methods exploit symbolic representations, still, the paper is written as two papers, i.e. with two separate methods and two separate experiments sections. This really reduces the potential impact of the paper.
2- The methods were compared against some baselines. However, in order to strongly support the main claim (use of symbols in human guidance brings efficiency), the methods should be compared with human guidance without symbols (e.g. TAMER+RL [17,18,19]). Without such comparison, it is not possible to convince and justify the advantage of the dual representation framework in tasks where human guidance is used.
3- The authors should drop the statements related to encoding/understanding "human internal representations", which is emphasized especially in the Introduction section. Some examples: Does aligning better with human internal state really correspond to encoding information that motivates human input (abstract)? Lines 31-33, related to understanding the internal representation of humans ([43-45] do not have any claim related to such understanding).

**Summary Of Recommendation:**

The paper is clearly written and works on an important topic, using dual representations to speed up learning (via minimizing the human guidance queries). The methods are also sound. However, the advantage of using symbols in learning with human guidance was not vigorously shown. Additionally, the central claim of the paper is not clear (which type of guidance should we use) and the methods and related experiments parts are not connected.

---

> ### Author Response · Authors · 2022-08-26
> **Response to Reviewer M5fV 1/2**
>
> Thank you for the constructive comments and helpful feedback.
>
> **RM5fV-Q1: Connections between the two methods.**
>
> RM5fV-A1: Our goal is to show the proposed framework is useful for more than one popular human-in-the-loop robot learning paradigm. There exists a connection between the two methods. Evaluative feedback (EF) and preference learning (PL) both elicit human evaluations for observed agent policies. EF targets state-action pairs which are fine-grained, while most PL targets full trajectories which are too coarse. Hence the former needs a grouping mechanism (for generalization, e.g., to estimate the feedback prediction error as in Eq. 2), and the latter needs a segmentation mechanism. Using abstract states is a unified approach to provide both. We have added this discussion to the revised manuscript in related work (revision, Line 72 and Line 80) and discussion (Line 277-280).
>
> **RM5fV-Q2: The methods should be compared with human guidance without symbols (e.g. TAMER+RL [17,18,19]).**
>
> RM5fV-A2: We would like to clarify that our EF-X% baselines are all TAMER+RL (TAMER+SAC), so we have demonstrated the effectiveness of the methods against human guidance without symbols. The only difference is that in TAMER+RL, humans choose when to provide feedback. This design causes a large variance in performance due to differences in human reaction time, expertise level, fatigue, etc. Therefore, EF-100% (stops every step and asks for feedback) can be seen as the performance upper bound of TAMER+RL, which was included in the simulation experiments. We further provide results for EF-100% in Placing-Real with actual humans (please see attached pdf). Our algorithm achieves comparable performance to the upper bound with less human feedback compared to TAMER+RL baselines in both simulation and with real humans.
>
> **RM5fV-Q3: Statements related to encoding/understanding "human internal representations".**
>
> RM5fV-A3: Thanks for the suggestion. We have updated the manuscript to remove these statements in the abstract, introduction, and related work.
>
> **RM5fV-Q4: Symbolic scene graphs are emphasized. However, what I see is a binary vector that represents the environment. The full power of the graph structure is not used (e.g. object-object relations are not used).**
>
> RM5fV-A4: Object-object relations are used, e.g., Luna Lander has a contact(leftLeg,ground) state. The agent does not have direct control over the legs, so they are considered objects. Nonetheless, most of the objects are static in our tasks so their relations are not used. We do agree that exploiting the full power of the graph structure is important in the future.
>
> **RM5fV-Q5: The claim on "humans adopting the slow system" in teaching.**
>
> RM5fV-A5: We agree that this is task-dependent. We have revised the original sentence in the following way: We hypothesize that the “slow” system and its representation are useful in guiding learning agents (revision, Line 42).
>
> **RM5fV-Q6: Lines 48-50: The correspondence between Evaluative Feedback and uncertainty awareness is not clear. The correspondence between Preference Learning and scene graph-based trajectory segmentation is not clear.**
>
> RM5fV-A6: Text updated to clarify these: We propose two novel algorithms, Dual Representation--based Evaluative Feedback (DREF) and Dual Representation--based Preference Learning (DRPL). DREF is based on the idea of uncertainty-aware active learning that allows the agent to estimate the uncertainty of human feedback in unseen states.
> DRPL builds upon scene graph--based trajectory segmentation and selection, which allows efficient reward learning from carefully chosen trajectory segments (revision, Line 47-51).

---

> > ### Author Response · Authors · 2022-08-26
> > **Response to Reviewer M5fV 2/2**
> >
> > **RM5fV-Q7: Related work**
> >
> > RM5fV-A7: Thanks for these references! We have included all suggested references in the revised version (Line 83-86).
> >
> > **RM5fV-Q8: Line 91: The tasks are probably represented at different levels also in humans.**
> >
> > RM5fV-A8: We have removed this sentence.
> >
> > **RM5fV-Q9: How "maximization of human feedback" is integrated into the loss in lines 131-132 is not clear.**
> >
> > RM5fV-A9: We follow the standard actor-critic framework, in which human feedback is interpreted as an additional Q-value function. The critics (human and environment) estimate the value functions. The actor updates the policy distribution in the direction suggested by the weighted average of both critics. We have updated Eq. 3 (now Eq. 3-5) to separate the loss for the actor and losses for two critics to clarify this (revision, Line 141-143).
> >
> > **RM5fV-Q10: Line 136 (and line 7 in Algo 1): It is not clear exactly which state is queried from the ones ranked smaller than k.**
> >
> > RM5fV-A10: Line 7 of Alg. 1 is used to determine whether we should query for human feedback in the current state s_t, and s_t belongs to an abstract state g_t. If g_t’s UCB value is ranked top k among all abstract states, we query human feedback in s_t.
> >
> > **RM5fV-Q11: Line 141: Statement related to being "meaningful" to humans.**
> >
> > RM5fV-A11: Removed “in a way that is meaningful to humans” (revision, Line 155).
> >
> > **RM5fV-Q12: Line 154: Related to the length of the trajectories: Why does your initial human insight idea not work here? No human would sum up any reward, they would simply say this is better than the other.**
> >
> > RM5fV-A12: For long-horizon tasks, trajectories are long and often involve several abstract state changes, some of them are good and some are bad. This makes it difficult to judge which trajectory is better overall. DRPL presents well-segmented trajectories to the humans so the preference judgment is easier hence reward learning is more efficient.
> >
> > **RM5fV-Q13: Is IRL implemented in Section 3.3? What are p and P (also in Algo 2)?**
> >
> > RM5fV-A13: Yes we use the Bayesian IRL algorithm in (Bıyık, & Sadigh, 2018; Bıyık, et al., 2021). P(q|theta) is the probability of human preference response q given the reward weights, and p(theta) is the probability distribution of reward weights. We have clarified this in the text (revision, Line 184) and Algorithm 2 (Line 2 & Line 7).
> >
> > **RM5fV-Q14: Line 209: "less effort": Effort was not measured. use "reported effort"**
> >
> > RM5fV-A14: We have revised this accordingly (revision, Line 231 and Fig.4 caption).
> >
> > **RM5fV-Q15: What is alignment score? Where is it defined? Was it not possible to provide the results using a common metric in Sections 4.1 and 4.2?**
> >
> > RM5fV-A15: It is not possible. In evaluative feedback, we train an RL agent to maximize the reward gained from the environment, so the cumulative reward is the performance metric. In preference learning, we estimate the unobserved reward function, so alignment to the true reward is used. The alignment score is a standard metric used in preference learning (Sadigh et al., 2017; Bıyık & Sadigh, 2018), which is the cosine similarity distance between the estimated reward weights and true weight (revision, Line 248).

---

> > ### Author Response · Authors · 2022-08-26
> > **EF-100% results with actual humans**
> >
> > For RM5fV-Q2.

---

### Meta-Review · Area_Chair_PBqW · 2022-08-13

**Recommendation:** Accept (Poster)
**Confidence:** 3

**Metareview:**

This paper proposes human-guided robot learning methods that require small number of feedbacks from humans. Two different algorithms, DREF and DRPL, are presented and evaluated with simulated and real human data and on simulated and real robots.

Strengths:
- The proposed algorithms demonstrate clear improvements in performance over the baselines.
- Efficiently linking low-level sensorimotor control and heigh-level human guidance.
- The number of experiments is adequate (both simulation and real experiments).

Weaknesses:
- Two different human guidance methods, DREF and DRPL, with almost no connection in between.
- Usefulness of the scene graphs is not thoroughly evaluated. Comparisons with other representations are required. Human experts need to pre-define the scene graphs.
- Disconnection between the paper’s idea / claim and the proposed evaluation.

---- Post rebuttal ----

During the rebuttal period, the authors successfully addressed some of the reviewers’ concerns by providing detailed explanations or removing inappropriate claims. However, the presentation quality of the paper needs to be further improved as one of the reviewers pointed out:
“(1) making the mathematical notation clearly connect the scene graph representation with the MDP representation (e.g., either via the state representation or the reward parameterization) and (2) somehow indicating the relationship between these dual representations clearly in the visuals and narrative. “




**Best Paper Nomination:**

No

---

> ### Author Response · Authors · 2022-08-26
> **Response to AC and all reviewers**
>
> We wholeheartedly thank you all for your constructive feedback. In addition to answering your individual questions, we have made the following changes to the manuscript to address several shared concerns. Please see the uploaded revision (with Appendix at the end) for the changes.
> - We have made the connections between the two proposed methods clear (please see RM5fV-A1, R8h9i-A2, RJYfb-A1).
> - We have explained why it is necessary to use human-defined representations, and why alternative representations such as learned ones are less desirable. We have also made the paper’s claim more consistent with the proposed evaluation (please see R8h9i-A1).
> - We have updated Eq. 3 to separate the loss for the actor and losses for two critics to clarify how the agent is trained to maximize the expected positive feedback from humans (please see RM5fV-A9, R32aJ-A3).
> - We have clarified the definition of “alignment score” which is used to evaluate preference learning algorithms (please see RM5fV-A15, RJYfb-A6).
> - We have included all the references suggested.
>
> Additionally, we made the following important changes to the manuscript to further improve our results:
> - For the real robot experiments with humans, we originally include results from three / two humans for each task (Reaching / Placing) in each paradigm (evaluative feedback and preference learning). We have added results from additional human subjects so we have six trials in each task and each paradigm. We have updated Fig. 4 and Fig. 6 in the text. Additional results further support the main conclusions of the experiments.
> - For preference learning in simulation and with humans, the results had large variances (Fig. 5, 6). We found that the issue can be significantly alleviated by 1) fixing an issue with the belief update MCMC sampling process in the Bayesian IRL algorithm we used [40] to ensure better convergence, and 2) making stimulus representation more friendly to humans so the position of the gripper is more clear in the real human studies. We have updated Fig 5 and 6 in the revised manuscript.

---

> > ### Author Response · Authors · 2022-08-26
> > **Revised manuscript**
> >
> > Please see attached pdf, thanks!